



**Atmospheric mercury in the southern hemisphere – Part 1: Trend and inter-**
**annual variations of atmospheric mercury at Cape Point, South Africa, in 2007**
**-2017, and on Amsterdam Island in 2012 - 2017**
Franz Slemr[1], Lynwill Martin[2], Casper Labuschagne[2], Thumeka Mokolo[2], Hélène Angot[3],
Olivier Magand[4], Aurélien Dommergue[4], Philippe Garat[5], Michel Ramonet[6], Johannes Bieser[7]
Corresponding author: Franz.Slemr@mpic.de
Second corresponding author: Lynwill.Martin@weathersa.co.za
[1]Max-Planck-Institut für Chemie (MPI), Air Chemistry Division, Hahn-Meitner-Weg 1, D-55128 Mainz,
Germany
[2]South African Weather Service c/o CSIR, P.O.Box 320, Stellenbosch 7599, South Africa
[3]Institute of Arctic and Alpine Research, University of Colorado Boulder, Boulder, CO, USA
[4]Institut des Géosciences de l'Environnement, Univ Grenoble Alpes, CNRS, IRD, Grenoble INP, 38400
Grenoble, France
[5]LJK, Univ Grenoble Alpes, CNRS, IRD, Grenoble INP, 38401 Grenoble, France
[6]Laboratoire des Sciences du Climat et de l'Environnement, LSCE-IPSL (CEA-CNRS-UVSQ), Université
Paris-Saclay, 91191 Gif-sur-Yvette, France
[7]Helmholtz-Zentrum Geesthacht (HZG), Institute of Coastal Research, Max-Planck-Str. 1, D-21502
Geesthacht, Germany




**Abstract**
The Minamata Convention on mercury (Hg) entered into force in 2017, committing its 116 parties (as
of January 2019) to curb anthropogenic emissions. Monitoring of atmospheric concentrations and
trends is an important part of the effectiveness evaluation of the Convention. A few years ago (in 2017)
we reported an increasing trend of atmospheric Hg concentrations at the Cape Point Global
Atmospheric Watch (GAW) station in South Africa (34°21′S, 18°29′E) for the 2007 – 2015 period. With





2 more years of measurements at Cape Point and the 2012 – 2017 data from Amsterdam Island
(37°48´S, 77°34´E) in the remote southern Indian Ocean, a more complex picture emerges: at Cape
Point the upward trend for the 2007 – 2017 period is still significant but none or slightly downward
trend was detected for the period 2012 – 2017 both at Cape Point and Amsterdam Island. The upward
trend at Cape Point is thus driven mainly by the 2007 - 2014 data. Using ancillary data on $^{222}$Rn, CO, $O_3$,
$CO_2$, and $CH_4$ from Cape Point and Amsterdam Island the possible reasons for the trend and its change
are investigated. In a companion paper this analysis is extended for the Cape Point station by
calculations of source and sink regions using backward trajectory analysis.
**1 Introduction**
Mercury (Hg) is an environmental toxicant emitted by both natural and anthropogenic sources – the
latter regulated by the Minamata Convention. This Convention, which entered into force in August
2017, requires periodic effectiveness evaluation (Article 22) to ensure that it meets its objectives. This
evaluation will be based on a combination of Hg monitoring data, including levels of Hg and Hg
compounds in air, biota, and humans. A few years ago, we reported an upward trend of atmospheric
mercury concentrations at the Cape Point Global Atmospheric Watch (GAW) station at Cape Point
(CPT, 34°21´S, 18°29´E) in South Africa for the 2007 – 2015 period (Martin et al., 2017). An upward
trend was surprising because manual mercury measurements at the same site in 1995 – 2004 showed
a downward trend. Downward trends of atmospheric mercury concentrations and of mercury wet
deposition have also been reported for many sites in the northern hemisphere (Temme et al., 2007;
Cole et al., 2014; Steffen et al., 2015; Weigelt et al., 2015; Weiss-Penzias et al., 2016; Marumoto et al.,
2019) but Cape Point has been the only station in the southern hemisphere with a long enough
mercury concentration record to calculate trends. The northern hemispheric downward trend has
been attributed to decreasing emissions from the North Atlantic Ocean due to decreasing mercury
concentrations in subsurface water (Soerensen et al., 2012) and more recently to decreasing global
anthropogenic emissions mainly due to the decline of mercury release from commercial products and
the changes of $Hg^0/Hg^{2+}$ speciation in flue gas of coal-fired utilities after implementation of NOx and
$SO_2$ emission controls (Zhang et al., 2016). Mercury uptake by terrestrial vegetation has also been
recently proposed to contribute to the downward trend (Jiskra et al., 2018).
In the meantime, mercury measurements at several other sites in the southern hemisphere have
become available (Sprovieri et al., 2016, 2017). Atmospheric mercury is quite uniformly distributed
throughout the southern hemisphere (Slemr et al., 2015) and its concentrations ($\sim$ 1.0 ng m$^{-3}$) are
substantially lower than those found at remote sites in the northern hemisphere ($\sim$1.5 ng m$^{-3}$)
(Sprovieri et al., 2016). Opposite to a pronounced seasonal variation with a maximum in early spring
and a minimum in autumn in the northern hemisphere (Sprovieri et al., 2016), hardly any seasonal



variation has been observed at Cape Point and Amsterdam Island (Slemr et al., 2015). The absence of
a pronounced seasonal variation in the southern hemisphere has been recently attributed to mercury
uptake by the terrestrial vegetation which, due to land distribution, is smaller in the southern
hemisphere (Jiskra et al., 2018).
In this paper we analyse the Cape Point (CPT) data for the 2007-2017 period and compare them with
the data from Amsterdam Island (AMS) obtained in the years 2012-2017. Mercury concentrations
remains nearly constant at both sites during the 2012 – 2017 period. Using simultaneously measured
$^{222}$Rn, CO, O$_3$, CO$_2$, and CH$_4$ concentrations at CPT and AMS we investigate the possible reasons for the
trend and its change.
**2 Experimental**
The Cape Point station (CPT, 34°21′S, 18°29′E) is located on the southern tip of the Cape Peninsula
within the Cape Point National Park at the summit of a 230 m a.s.l. peak about 60 km south of Cape
Town. The site is operated as one of the Global Atmospheric Watch (GAW) baseline monitoring
observatories of the World Meteorological Organisation (WMO) by South African Weather Service and
its current continuous measurements include Hg, CO, O$_3$, CH$_4$, CO$_2$, $^{222}$Rn, N$_2$O, several halocarbons,
particles, and meteorological parameters (Martin et al., 2017).
Amsterdam Island (AMS, 37°48′S, 77°34′E) is a small island (55 km$^2$) in the southern Indian Ocean,
3400 km and 5000 km downwind of Madagascar and South Africa, respectively. The station is located
at Pointe Bénédicte, at the northwest end of the island at an altitude of 55m a.s.l. Labelled GAW/WMO
Global site, the Amsterdam site hosts instruments occurring in the framework of the French national
observation service named ICOS-France-Atmosphere as well as the Global Observation System for
Mercury (GOS4M), for long-term monitoring of greenhouse gases and mercury species, respectively.
The site is ensured by the administration of Terres Australes and Antarctiques Françaises (TAAF), the
French Southern and Antarctic Lands, and scientifically operated by the French Polar Institute (IPEV).
Currently, CO, O$_3$, CO$_2$, CH$_4$, $^{222}$Rn, total aerosol number, carbonaceous aerosol, and meteorological
parameters are continuously monitored at the site (Angot et al., 2014).
Atmospheric mercury has been measured since March 2007 at CPT and since January 2012 at AMS
using Tekran 2537 (Tekran Inc., Toronto, Canada) at both sites. The instruments are based on mercury
enrichment on a gold cartridge, followed by a thermal desorption and a detection by cold vapour
atomic fluorescence spectroscopy (CVAFS). Switching between two cartridges allows for alternating
sampling and desorption and thus results in a full temporal coverage of the mercury measurement.
The instruments are automatically calibrated every 25 h at CPT and every 69 h at AMS using internal
mercury permeation sources which in turn were annually checked by manual injections of saturated


Hg vapour from a temperature-controlled vessel. To ensure the comparability of the mercury
measurements, Tekran instruments at both sites have been operated according to the Global Mercury
Observation System (GMOS) standard operating procedures (SOP, Munthe et al., 2011).
The instrument at CPT has been operated with 15 min resolution since March 2007. At AMS, the Tekran
speciation unit (Tekran 1130 and 1135) coupled to the Tekran 2537B analyser (Tekran Inc. Toronto,
Canada) was in operation since January 2012 until December 10, 2015. Gaseous elemental mercury
(GEM) was measured with 5 min resolution during this period. Concentrations of gaseous oxidized
(GOM) and particulate mercury (PM) were below the detection limit for most of the time (Angot et al.,
2014). Consequently, only GEM has been continuously measured with Tekran 2537A/B analyser since
December 14, 2015, with a resolution of 15 min as at the Cape Point while GOM and PM species
continued to be collected on CEM filters on weekly frequencies.
With GEM concentrations of ~ 1 ng m$^{-3}$ and a sampling flow rate of 1l (STP) min$^{-1}$ mercury loads on gold
cartridges are ~ 5 pg and ~ 15 pg with 5 min and 15 min long sampling, respectively. A measurement
bias with loads <10 ng m$^{-3}$ due to internal Tekran integration procedure (Swartzendruber et al., 2009;
Slemr et al., 2016a; Ambrose, 2017) can impair comparability of the measurements made with 5 min
resolution with those made with 15 min resolution. The possible bias of the measurements at AMS in
2012-2015 was eliminated by optimising the integration parameters (Swartzendruber et al., 2009). The
absence of bias was shown by calculating the monthly variation coefficients of the 5 and the 15 min
measurements at AMS. The average monthly variation coefficients were 5.81 ± 2.15 % (n=48) and 5.83
± 1.48 % (n=24) for 5 min and 15 min resolution, respectively, and they are statistically not
distinguishable. We thus conclude that the measurements at AMS with 5 min resolution are
comparable to those with 15 min.
**3 Results and discussion**
3.1 Seasonal variation
Figure 1 shows seasonal GEM variations at CPT (upper panel) and AMS (lower panel). They were
calculated by averaging of monthly medians over the period of 2012 – 2017. Similar plots were
obtained by averaging of monthly averages in the same period. The amplitude of the seasonal variation
at AMS is with > 0.1 ng m$^{-3}$ somewhat larger than at CPT (~ 0.08 ng m$^{-3}$). The standard deviations of
monthly average concentrations are larger at CPT than at AMS indicating higher interannual variation
at CPT. Smaller standard deviations at AMS enable to detect significant differences between the
months with the highest (June, July, and August) and the lowest three (November, February, and
October) GEM concentrations. GEM concentrations in December and January lie outside of an



otherwise nearly sinusoidal seasonal variation but their differences to GEM averages in other months
are not significant. No significant differences between monthly averages at CPT were found.
In summary, maximum GEM concentrations at AMS are observed in austral winter (June – August) and
the lowest GEM concentrations in austral summer. Austral winter is the season with the most frequent
fast transport from southern Africa to AMS (June – October; Miller et al., 1993) coinciding also with
maximum $^{222}$Rn concentrations at AMS (May – August) as another indicator of continental influence
(Polian et al., 1986). The most frequent events at AMS in 1996 – 1997 with high CO mixing ratios
occurred also in austral winter (June – October, Gros et al., 1999). Biomass burning in southern Africa
peaks in austral winter (July – October, Duncan et al., 2003) and we therefore conclude, in agreement
with Angot et al. (2014), that mercury from biomass burning in southern Africa combined with its fast
transport to AMS is mostly responsible for the seasonal variation observed there. Reduced uptake of
atmospheric GEM by terrestrial biomass of southern Africa in austral winter (Jiskra et al., 2018) can
also contribute.
3.2 Trends at CPT in 2007 - 2017
Figure 2 shows annual median GEM concentrations at CPT (2007 – 2017) and at AMS (2012 – 2017).
Table 1 shows the trends of GEM, $CO_2$, $^{222}$Rn, CO, $CH_4$, and $O_3$ at CPT in the 2007-2017 period as
calculated by least square fit of monthly averages or medians (medians are shown in Figure 1 of
Supporting Information). Monthly average and median GEM concentrations show a significant upward
trend of 7.69 ± 2.11 and 7.01 ± 2.11 pg m$^{-3}$ yr$^{-1}$, respectively. The upward trends of $CO_2$ (2.07 ± 0.03
ppm yr$^{-1}$ for averages and 2.08 ±0.02 ppm yr$^{-1}$ for medians) and $CH_4$ (5.70 ± 0.66 ppb yr$^{-1}$ for averages
and 5.85 ±0.53 ppb yr$^{-1}$ for medians) are comparable to worldwide trends of 2.24 ppm yr$^{-1}$ for $CO_2$ and
6.9 ppb yr$^{-1}$ for $CH_4$ in 2008-2017 (WMO Greenhouse Gas Bulletin, 2018) . For the interpretation of the
GEM trend, the most revealing is the non-significant trend in $^{222}$Rn and the significant downward trend
in CO. $^{222}$Rn is a radioactive gas of predominantly terrestrial origin with a half-life of 3.8 days. Non-
significant $^{222}$Rn trend thus implies a nearly constant ratio of oceanic to continental air masses over
the 2007 – 2017 period and rules out larger shifts in climatology of CPT as the cause of the observed
GEM trend. Biomass burning is a major source of CO in the southern hemisphere (Duncan et al., 2003;
Pirrone et al., 2010) and at the same time a major source of Hg (Friedli et al., 2009). The downward
trend of CO thus rules out increasing Hg emissions from biomass burning to be responsible for the
upward GEM trend at CPT.  The downward trend of CO at CPT is consistent with the decreasing CO
emissions in 2001 – 2015 (Jiang et al., 2017). They report decreasing CO emissions from biomass
burning from boreal North America, boreal Asia and South America and no change in Africa.
3.3 Trends at CPT and AMS in 2012 - 2017



Monthly GEM averages and medians at AMS and CPT in the 2012 - 2017 period are not statistically
distinguishable according to the paired student t test. Monthly $CO_2$ averages at CPT are significantly
higher than at AMS (at >99.9% significance level) but medians cannot be distinguished. Medians of
$CO_2$, $^{222}Rn$, CO, and $CH_4$ are less influenced by occasional events with extremely high values and as such
tend to be smaller than averages. Because such events are less frequent at AMS than at CPT, the
differences between monthly averages and medians are always higher at CPT than at AMS. This
explains why the $CO_2$ monthly averages are significantly higher at CPT than at AMS but the medians
are not. Similarly, the significance of the monthly differences between higher CO at CPT and lower at
AMS is >99.9% for averages but only >99% for medians. Monthly $CH_4$ mixing ratios are always higher
at CPT than at AMS with >99.9% significance both for averages and medians. The most pronounced
difference between CPT and AMS is in $^{222}Rn$ concentrations: monthly averages and medians at CPT are
on average 16.6 and 12.6 times higher, respectively, than at AMS. In summary, higher monthly $CO_2$,
CO, $CH_4$, and especially $^{222}Rn$ averages and medians at CPT than at AMS clearly demonstrate higher
influence of continental air masses at CPT because all these species are predominantly of terrestrial
origin. Statistically comparable GEM concentrations at AMS and CPT in 2012 – 2017, on the contrary,
suggest that terrestrial GEM sources do not play a major role at CPT. This conclusion is supported by
an analysis of GEM/$^{222}Rn$ ratios in events with enhanced $^{222}Rn$ concentrations observed at CPT (Slemr
et al., 2013) which found terrestrial surface of southern Africa to be rather a sink of GEM than a source.
This is further discussed in the companion paper (Bieser et al., 2019).
Tables 2 and 3 shows the 2012 – 2017 trends of GEM, $CO_2$, $^{222}Rn$, CO, and $CH_4$ at AMS and CPT,
respectively. The AMS monthly average and median GEM concentrations do not show any significant
trend. At CPT monthly average GEM concentrations do not show any significant trend, whereas median
GEM concentrations show a significant slight downward trend (at >95% significance level). As in the
2007-2017 period the neutral to slightly downward GEM trend at CPT is accompanied by no significant
trend in $^{222}Rn$. Opposite to the 2007 – 2017 period CO does not show any significant downward trend
whereas $O_3$ (not listed) shows a small significant upward trend in monthly averages but not in monthly
medians.
An inspection of Figure 2 shows that the GEM trend at CPT in 2007 – 2017 period is driven mainly by
the 2009 – 2014 period. Table 1 of supporting information (SI) shows the trends of GEM, $^{222}Rn$, CO,
$CH_4$, and $O_3$ at CPT for the 2007 – 2014 period. Monthly average and median GEM concentrations
increased by 16.91 ± 3.60 and 16.18 ± 3.61 pg m$^{-3}$ yr$^{-1}$, respectively. This upward GEM trend is
accompanied by no trend in $^{222}Rn$ and $O_3$, and small downward trend in monthly average CO mixing
ratios but not in medians.


In summary, the 2007 – 2017 time series of GEM concentrations at CPT consists of two parts: one
starting in 2007 and ending approximately in 2014 with a pronounced upward trend and the other
without any or even slightly downward trend starting in 2012. The absence GEM trend in 2012 – 2017
at CPT is in agreement with the absence of the GEM trend at AMS in the same period. The upward
trend thus appears to have changed between 2012 and 2014. The absence of $^{222}$Rn trends at CPT for
2007 – 2017 and the subperiods 2007 – 2014 and 2012 - 2017 points to nearly constant ratio of marine
and continental air masses over the years and thus rules out shifts in regional climatology being
responsible for the GEM trends. A downward trend of CO over the 2007 – 2017 period and none or
just significantly downward one for the subperiods 2007 – 2014 and 2012 – 2017 makes it unlikely that
increasing Hg emissions from biomass burning could be the reason for upward trend of GEM
concentrations at CPT. We note that both $^{222}$Rn concentrations and CO mixing ratios have a very
pronounced seasonal variations which make it difficult to determine significant trends over shorter
periods.
3.4 Inter-annual variations of GEM concentrations
A plot of annual median GEM concentrations in Figure 2 (annual averages provide a very similar pattern
and are not shown) shows that median concentrations in 2007 and 2008 are only slightly lower than
in 2015 - 2017. It is the steady increase from the lowest GEM concentrations in 2009 to the highest
ones in 2014 at CPT (the latter 2$^{nd}$ highest at AMS in 2012 – 2017 period) which seems to be responsible
for the upward trend in 2007 – 2017 at CPT and no trend for 2012 -2017 period for both CPT and AMS.
Exceptionally low annual GEM concentrations in 2009 (average and median of 0.918 and 0.913 ng m$^-$
$^3$, respectively) and exceptionally high ones in 2014 (average and median of 1.090 and 1.094 ng m$^{-3}$,
respectively, at CPT, 1.050 and 1.053 ng m$^{-3}$, respectively, at AMS) seem to be a near global
phenomenon. The years 2009 and 2014 show the largest deviations (a negative one in 2009, a positive
one in 2014) from the linear 2000 - 2014 trend of annual GEM average concentrations recorded at 18
sites in North America (Figure 8 b of Streets et al., 2019). At Mace Head, a site in Ireland, GEM annual
average and median concentrations in 2009 were the lowest over the 1996 – 2013 period (supporting
Information of Weigelt et al., 2015). The reasons for these near global inter-annual variations are not
clear. Global anthropogenic Hg emissions do not vary much from year to year (mostly by less than 5%)
and have been steadily increasing over the 2010 – 2015 period (Streets et al., 2019). Between 2000
and 2010 they steadily increased by ~10% (Streets et al., 2017 and 2019). These emission estimates do
not include Hg from biomass burning but CO emissions from biomass burning, as a proxy for Hg
emissions, were somewhat lower in 2008 and 2009 but not exceptionally high in 2014 (Jiang et al.,
2017). Annual volcanic SO$_2$ emissions, as a proxy for volcanic Hg emissions, also do not show
exceptionally low emissions in 2009, although the emissions in 2014 were the second highest (after


2011) on record in the 1996 – 2018 period (https://disc.gsfc.nasa.gov/datasets/MSVOLSO2L4_V-
3/summary).
Tropospheric mercury concentrations were found to be influenced by El Niño Southern Oscillation
(ENSO) (Slemr et al., 2016b). Such influence could also be a reason for the observed inter-annual
variation of GEM concentrations at Cape Point. Table 4 shows correlations of 3 months running
averages and medians of GEM concentration at CPT with 3 months running average of Southern
Oscillation Index (SOI) for 2007 – 2014 and 2012 – 2017 and compares them with the 2012 – 2017
period at AMS. 3 month running averages and medians were taken instead of monthly averages to
take account for time of intra-hemispheric mixing. Correlations of CO mixing ratios with SOI
(www.cpc.ncep.noaa.gov/data/indices/soi.3m.txt) at CPT for 2007 – 2014 and 2012 – 2017 are also
listed. CO vs SOI correlations for AMS were not made because the CO mixing ratios are available only
since December 2015 until December 2017.

Table 4 shows negative correlations of GEM concentrations with SOI at AMS for 2012 - 2017 with a lag
of 6-8 months both for averages and medians. Relative GEM (after detrending) at CPT also
anticorrelates with SOI at CPT in the 2007 – 2014 period as does CO mixing ratio (deseasonalised) in
the same period, both with a slightly longer lag of 9 – 11 months. Anticorrelations of GEM
concentrations and CO mixing ratios with SOI with similar lags were reported by Slemr et al. (2016b)
who interpreted them as a sign for biomass burning being the driving force for the inter-annual
variation of GEM and CO. The GEM and CO vs SOI correlations for the 2012 – 2017 period at CPT are
both positive and the CO vs SOI correlation is significant only at >95% level. For the 2007 – 2017 period
at CPT, encompassing both periods, also a negative correlation of GEM vs SOI was found but with a
lower significance level of only >95%. The different correlations of GEM and CO with SOI at CPT for the
period 2012 – 2017 from those at CPT in 2007 – 2014 and of GEM vs SOI at AMS in 2012 – 2017 clearly
shows that at least at CPT the mechanism for inter-annual variations changed.
Correlations of detrended monthly GEM averages and medians at CPT with North Atlantic Oscillation
(NAO)                                                                                                    index
(www.cpc.ncep.noaa.gov/products/precip/Cwlink/pna/norm.nao.monthly.b5001.current.ascii.table)
over the period 2007 – 2017 were not significant for medians and just significant (>95%) for averages
with a lag of 11 months. In the 2012 – 2017 period the correlations of GEM with NAO index were
significant (>95%) with a delay of 0- and 8-months both for monthly medians and averages (both not
detrended). The correlation with 0 months delay is negative and that with 8-month delay is positive.
At AMS monthly GEM averages correlate with NAO index with a delay of 3, 5, and 6 months, all at a
significance level of >95%. Monthly medians correlate with a delay of 5 and 6 months, the latter even



at a significance level of > 99%. In summary, there seems to be some influence of NAO on GEM
concentration. The influence is more pronounced at AMS than at CPT, probably because of more
regional influence at the latter site.
The annual GEM minimum in 2009 and the maxima in 2014 and 2012 at CPT as well as the annual
minima in 2015 and 2017 and maxima in 2014 and 2016 at AMS fit a biennial tendency already
mentioned by Martin et al. (2017) with mostly lower annual GEM concentrations in odd years and
higher ones in even years. The biennial tendency is also apparent in the annual median and average
CO mixing ratios at CPT (there are only two years with CO measurements at AMS), with mostly lower
values in odd years and higher ones in even years, similar to GEM concentrations. Meehl and Arblaster
(2001, 2002) note a relation between Tropospheric Biennial Oscillation (TBO) and ENSO, the latter also
with a biennial tendency.
In summary, a part of the inter-annual variation of GEM concentrations seems to be related to
teleconnections like ENSO, TBO and NAO.
**Conclusions**
Martin et al. (2017) reported an upward trend of GEM concentrations at CPT from March 2007 to June
2015. With two and a half year of more measurements at CPT until December 2017 and GEM
measurements at AMS since February 2012 until December 2017 a more complex picture emerged:
No significant trend of GEM concentrations was found at CPT and AMS for the period of AMS
measurements, i.e. 2012 – 2017. Upward trend of GEM concentrations at CPT in 2007 – 2015 reported
by Martin et al. (2017) is driven mainly by the 2009 – 2014 data with a minimum in 2009 and maxima
in 2012 and 2014. The latter two years with high annual GEM concentrations seem to be the reason
for absent trend in 2012 – 2017 period, although the upward trend over the whole 2007 – 2017 period
at CPT is still significant. A minimum of GEM concentrations in 2009 was also reported for stations in
North America and at Mace Head, Ireland. In addition, annual average and median GEM concentrations
at CPT and AMS show a biennial pattern with lower concentrations in odd years and higher ones in
even years. Because of the pronounced inter-annual variations, the calculated GEM trends will depend
on the year when the observations start and end and increasingly so, the shorter the observation
period is.
No trend was found in $^{222}$Rn concentrations and a slight downward trend in CO mixing ratios were
found at CPT in 2007 – 2017. Changing ratios of marine and continental air masses at CPT as well as
increasing mercury emissions from biomass burning can, therefore, be ruled out as the cause of the
upward GEM trend at CPT.



Monthly average GEM concentrations at CPT and AMS in 2012 – 2017 are statistically indistinguishable
while concentrations of species of terrestrial origin such as $CO_2$, $CH_4$, CO, and especially of $^{222}Rn$ clearly
show substantially higher values at CPT in comparison with those at AMS. Comparable GEM
concentrations at CPT and AMS despite much higher influence of terrestrial air masses at CPT thus
indicate that terrestrial GEM sources are of minor importance and the oceanic GEM sources are
dominating at CPT. This major conclusion will be substantiated by a companion paper in which the
GEM concentration will be, with help of backward trajectories, attributed to different source and sink
regions.

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



**Tables**

*Table 1. Trends at Cape Point for the 2007 – 2017 period. Calculated by LSQF from monthly averages and medians.*

| Species | Monthly | Annual slope | Unit | R, n, significance |
|---|---|---|---|---|
| GEM | average | 7.69 ± 2.11 | $pg\ m^{-3}\ yr^{-1}$ | 0.3098, 127, >99.9% |
| | median | 7.01 ± 2.11 | | 0.2846, 127, >99% |
| $CO_2$ | average | 2.208 ± 0.018 | $ppm\ yr^{-1}$ | 0.9955, 132, >99.9% |
| | median | 2.219 ± 0.017 | | 0.9964, 132, >99.9% |
| Rn | average | -0.76 ± 7.96 | $mBq\ m^{-3}\ yr^{-1}$ | -0.0085, 130, ns |
| | median | 0.05 ± 4.58 | | 0.0009, 130, ns |
| CO | average | -1.020 ± 0.301 | $ppb\ yr^{-1}$ | -0.2848, 132, >99% |
| | median | -0.503 ± 0.223 | | -0.1939, 132, >95% |
| $CH_4$ | average | 6.650 ± 0.402 | $ppb\ yr^{-1}$ | 0.8236, 132, >99.9% |
| | median | 6.895 ± 0.335 | | 0.8751, 132, >99.9% |
| $O_3$ | average | 0.263 ± 0.151 | $ppb\ yr^{-1}$ | 0.1510, 131, ns |
| | median | 0.260 ± 0.161 | | 0.1408, 131, ns |





*Table 2: Trends at Amsterdam Island for the 2012 - 2017 period.*

| Species | Monthly | Annual slope | Unit | R, n, significance |
|---|---|---|---|---|
| GEM | average | 4.10 ± 3.65 | pg m$^{-3}$ yr$^{-1}$ | 0.1371, 68, ns |
|  | median | 5.57 ± 3.61 |  | 0.1865, 68, ns |
| $CO_2$ | average | 2.487 ± 0.025 | ppm yr-1 | 0.9962, 72, >99.9% |
|  | median | 2.487 ± 0.026 |  | 0.9959, 72, >99.9% |
| Rn | average | -1.626 ± 1.018 | mBq m$^{-3}$ yr$^{-1}$ | -0.190, 70, ns |
|  | median | -0.557 ± 0.604 |  | -0.111, 70, ns |
| CO | average | -1.530 ± 2.405 | ppb yr$^{-1}$ | -0.131, 25, ns |
|  | median | -1.460 ± 2.351 |  | -0.128, 25, ns |
| $CH_4$ | average | 8.575 ± 0.786 | ppb yr$^{-1}$ | 0.7932, 72, >99.9% |
|  | median | 8.555 ± 0.793 |  | 0.7899, 72, >99.9% |


*Table 3. Trends at Cape Point for the 2012 – 2017 period.*

| Species | Monthly | Annual slope | Unit | R, n, significance |
|---|---|---|---|---|
| GEM | average | -8.65 ± 4.63 | pg m$^{-3}$ yr$^{-1}$ | -0.2211, 70, ns |
|  | median | -9.31 ± 4.55 |  | -0.2409, 70, >95% |
| $CO_2$ | average | 2.459 ± 0.035 | ppm yr$^{-1}$ | 0.9931, 72, >99.9% |
|  | median | 2.466 ± 0.030 |  | 0.9949, 72, >99.9% |
| Rn | average | 20.05 ± 18.87 | mBq m$^{-3}$ yr$^{-1}$ | 0.1269, 71, ns |
|  | median | 15.36 ± 10.51 |  | 0.1732, 71, ns |
| CO | average | -0.151 ± 0.692 | ppb yr$^{-1}$ | -0.0260, 72, ns |
|  | median | 0.053 ± 0.540 |  | 0.0117, 72, ns |
| $CH_4$ | average | 9.160 ± 0.979 | ppb yr$^{-1}$ | 0.7455, 72, >99.9% |
|  | median | 9.498 ± 0.818 |  | 0.8111, 72, >99.9% |




Table 4: Correlation of 3 months running average and median GEM concentrations and CO
mixing ratios with 3 months running average of SOI
(www.cpc.ncep.noaa.gov/data/indices/soi.3m.txt.  The CPT GEM data for 2007 – 2014 were
detrended, the CPT CO data for 2007-2014 and 2012-2017 deseasonalized using the average
monthly averages or medians over the period. No CO correlation is presented for AMS
because CO data are available only since December 2015 until December 2017. The delay
given in the last column is the one with the highest R. The delays in the brackets are
significant correlations with the second and third highest R.

| Site and period | | Equation | R, n, signif. | GEM delay [month] |
|---|---|---|---|---|
| AMS, GEM, 2012–2017 | average | GEM=-0.0227*SOI+1.0375 | -0.4145, 70, >99.9% | 7 (6-8) |
| | median | GEM=-0.0230*SOI+1.0390 | -0.4150, 70, >99.9% | 7 (6-8) |
| CPT, GEM, 2007-2014 | average | relGEM=-0.0330*SOI+1.0179 | -0.4554, 95, >99.9% | 10 (9-11) |
| | median | relGEM=-0.0373*SOI+1.0202 | -0.4934, 95, >99% | 10 (9-11) |
| CPT, CO, 2007-2014 | average | relCO=-0.0367*SOI+1.0199 | -0.4171, 95, >99.9% | 10 (9-11) |
| | median | relCO=-0.0340*SOI+1.0184 | -0.5406, 95, >99.9% | 10 (9-11) |
| CPT, GEM, 2012-2017 | average | GEM=0.0318*SOI+1.0371 | 0.4523, 69, >99.9% | 8 (7-9) |
| | median | GEM=0.0279*SOI+1.0385 | 0.3906, 69, >99.9% | 7 (7-9) |
| CPT, CO, 2012-2017 | average | relCO=0.0173*SOI+0.9995 | 0.2358, 71, >95% | 8 (9) |
| | median | relCO=0.0196*SOI+0.9991 | 0.2914,71, >95% | 9 (10-11) |





**Figures**

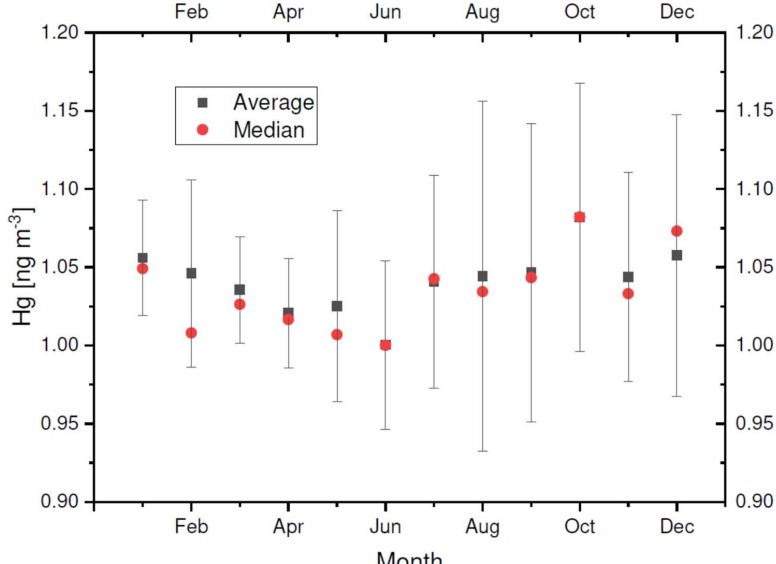


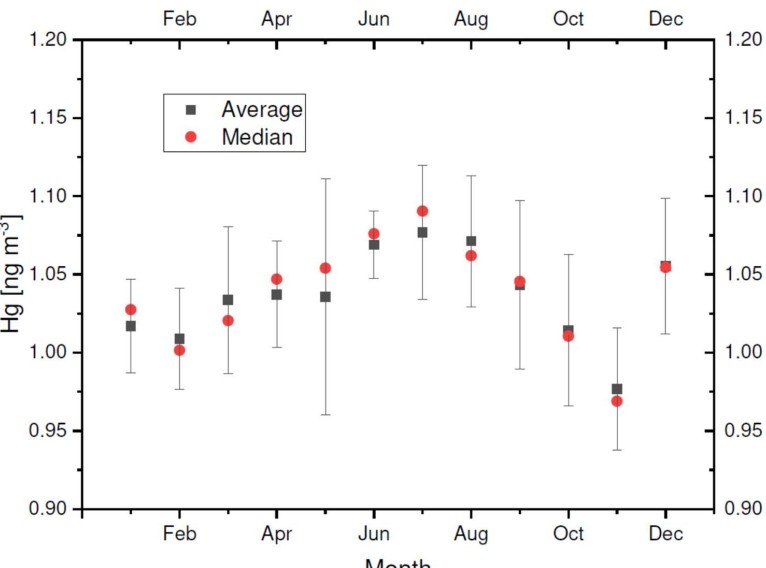


Figure 1: Seasonal variation of GEM in 2012 − 2017 at CPT (upper panel) and AMS (lower
panel). The points represent averages and medians of monthly medians over the 2012 − 2017
period. The bars represent the standard deviations of the monthly averages.




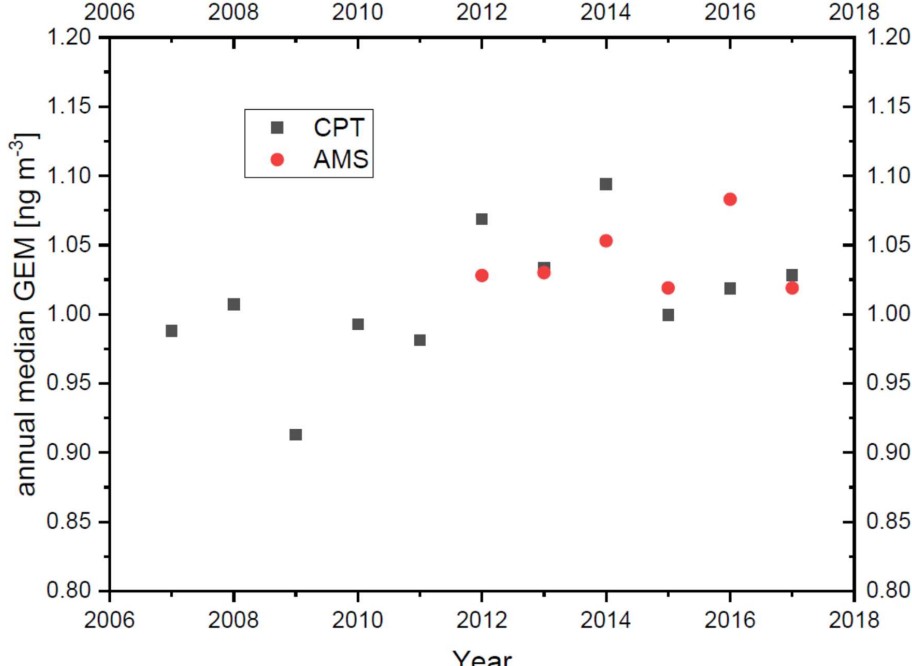



Figure 2: Annual median GEM concentrations at Cape Point (CPT) since March 2007 until
December 2017 and at Amsterdam Island (AMS) since February 2012 until December 2017.