# Peer review of "Supporting information"

_Atmospheric Chemistry and Physics, 2020_

## Referee Comment (RC1) · Anonymous Referee #1 · 25 Feb 2020

The manuscript addresses atmospheric mercury concentration changes and trends. No new concept, method or insight in time series/trend evaluation are presented. The authors based on the relation with 222Rn, CO, O3, and CH4, exploit the possible reasons for the trend change. A more ambitious goal could be target assessing the strength of each one of these species on mercury concentration changes applying probabilistic mass function in rotational matrix factorization. However, the paper can be relevant for mercury source assess, which maybe has potential implications for

policy abatement strategies afterward and also potential scientific contribution on the atmospheric mercury cycle (source and fade) understanding. The paper is well written, easer reading, discussed with expertise and I recommend publication in ACP especial issue.Âǎ Other comments: Line 36: I am afraid calling this trend, seems more annual comparison. Trend evaluation for so short period could be misleading since it can be affected by seasonality (for example, starting the time series in winter and finishing in autumn or starting summer/spring and finish in winter/autumn). It also can be affected by starting or finishing the time series in an El Niño year. Line 54-56: decrement of North Atlantic Ocean emission is rather a hypothesis than a scientific statement. Line 119-120: confidence level missed. Line 248: Anticorrelations should be replaced by negative correlation or inverse correlation. Figure 2 should be replaced for a more informative time series decomposition plot, presenting trend seasonal and random variable in an hour or daily (or at least monthly) time resolution. Such graphic can improves data exhibition, facilitate reader evaluation and can be easily calculated and plotted included by using open libraries for R and Python. Furthermore, the annual median or average is not suitable for trend evaluation since it damp variance and constrain significance.

---

## Referee Comment (RC2) · Anonymous Referee #2 · 19 Mar 2020

This study presents the long-term trend of atmospheric Hg at two monitoring sites in the Southern Hemisphere and explains the factors related to the trends. To my knowledge, this is an unique study monitoring the long-term trend of atmospheric Hg in the Southern Hemisphere. The finding of this study that the continuous increasing trend of atmospheric Hg concentrations in the Southern Hemisphere has been reversed (or weakened) since 2012-2014 is exciting to me. This is in contrast with some recent global bottom-up emission inventories that showed increasing anthropogenic Hg emis-

sions during 2010-2015, suggesting that the measures taken during implementation of Minamata Convention may have reduce the anthropogenic emissions in recent years. Therefore, this study is valuable to atmospheric research communities and government. This paper is well organized and easily to be read. I have several comments that I wish could be addressed by the authors before the final publication of this paper. 1. line 34-37: this seems to be confusing. I would suggest the authors to separate these two periods throughout the manuscript because the trends during these two period is quite different. As mentioned by the authors, the increasing trend was only observed between 2007 and 2014, and a declaration of an upward trend to 2017, to me, is at least not precise. If we look at the median (or mean) between 2007-2008 and 2016-2017, there is no clear increase of Hg concentrations. 2. line 57-59: the study by Zhang et al., 2016 simulated the trend between 1990 and 2010 and compared them with field observations, that is not study related to the trend in last ten years. Also, I did not see any remarks from this study that anthropogenic Hg emissions is decreasing. 3. line 77-82: I would suggest to add Figure 1 to show the locations of the sampling sites. In some of following sections, the authors discussed the effect of atmospheric transport patterns on seasonal Hg trend. It will be excellent that the authors could provide the major air mass origins in wet and dry seasons. 4. Line 133-144: From Figure 1, I see the seasonal variation in Hg at CPT (although not statistical significant) is opposite to that at AMS, with Hg peak in austral summer at CPT and in austral winter at AMS. I think the authors have not provide clear explanations for this. The contrast seasonal pattern indicate that the atmospheric transformation, foliar uptake, or oceanic emissions might play an unimportant role here. The authors suggest that the frequent biomass burning in southern African and prevailing transport in austral winter may explain the seasonal trend at AMS. My question is why the enhanced biomass burning activities in southern African did not cause an increase of Hg in austral winter at CPT? which is located more close to southern African continent. Is the air mass transport from southern African to CPT less in austral winter than summer? 5. line 182-183: here the authors should indicate the sources of GEM. Are the sources related to oceanic emissions or

long range transport from other countries or continents. This should be clarified here. 6. line 200-201: here the author declare no clear GEM trend in 2012-2017. But in many other sections the authors also say a slightly downward trend. I think the authors should provide a consistent conclusion for the trend during this period. 7. line 277-279: it is very interesting that the authors explore the relationship between GEM trend and climate change (e.g., ENSO). But I am wondering that how would the climate would effect the long-term trend of Hg. Will the occurrence of ENSO decrease or increase the natural Hg emissions from continents or oceans? I think it is better to provide a further discussion for this. Table 1-3: I think it is better to add the trend in percentile (the authors only provide absolute concentration trend) to these tables. This is because we can not read whether the trend is strong or not based on concentrations. For example, does the annual increase of 0.05 mBq m-3 yr-1 of median Rn concentrations indicate a strong increase for Rn concentrations? Figure 2: I would suggest the authors add a figure to show the means of Hg. the authors used the means many time throughout the paper.

―――――――――――――――――――

---

## Author Comment (AC1) · 14 May 2020

**Response to Anonymous Referee #1**

The manuscript addresses atmospheric mercury concentration changes and trends. No new concept, method or insight in time series/trend evaluation are presented. The authors based on the relation with $^{222}Rn$, CO, $O_3$, and $CH_4$, exploit the possible reasons for the trend change. A more ambitious goal could be target assessing the strength of each one of these species on mercury concentration changes applying probabilistic mass function in rotational matrix factorization.

*We show in this and the companion paper that mercury is unique in that its sources are predominantly oceanic whereas the sources of all other investigated species are predominantly terrestrial. We think that a more elaborated statistical treatment without a specific tracer for oceanic emissions here and in the companion paper would only confirm this finding without any additional insight.*

Line 36: I am afraid calling this trend, seems more annual comparison. Trend evaluation for so short period could be misleading since it can be affected by seasonality (for example, starting the time series in winter and finishing in autumn or starting summer/spring and finish in winter/autumn). It also can be affected by starting or finishing the time series in an El Niño year.

*We agree that trend estimations for short periods may be misleading. With the exception of 2007 at Cape Point (measurement since March) the measurements covered whole years. The trends for the 2012 – 2017 period were calculated mainly for comparison of Cape Point with Amsterdam Island. Both should be affected by El Niño and most of the other teleconnections in the same way.*

Line 54-56: decrement of North Atlantic Ocean emission is rather a hypothesis than a scientific statement.

*All the works cited in this and following sentence are only hypotheses with a varying degree of evidence in their support. We think that there is no monocausal explanation of the observations and would thus hesitate to consider the other works as scientific statements too.*

Line 119-120: confidence level missed.

*Calculated t of 0.04092 for 70 degrees of freedom is an order of magnitude smaller than P of 0.5 (t = 0.678).*

Line 248: Anticorrelations should be replaced by negative correlation or inverse correlation.

*Done.*

Figure 2 should be replaced for a more informative time series decomposition plot, presenting trend seasonal and random variable in an hour or daily (or at least monthly) time resolution. Such graphic can improve data exhibition, facilitate reader evaluation and can be easily calculated and plotted included by using open libraries for R and Python. Furthermore, the annual median or average is not suitable for trend evaluation since it damp variance and constrain significance.

*In the captions of the Tables 2 and 3 we now note that the trends were calculated from monthly averages and medians. Monthly median concentrations at CPT and AMS are shown in Figure SI1. Because of the discussion of interannual variation we prefer to plot annual values in the paper.*

---

## Author Comment (AC2) · 14 May 2020

**Response to Anonymous Referee #2**

This study presents the long-term trend of atmospheric Hg at two monitoring sites in the Southern Hemisphere and explains the factors related to the trends. To my knowledge, this is a unique study monitoring the long-term trend of atmospheric Hg in the Southern Hemisphere. The finding of this study that the continuous increasing trend of atmospheric Hg concentrations in the Southern Hemisphere has been reversed (or weakened) since 2012-2014 is exciting to me. This is in contrast with some recent global bottom-up emission inventories that showed increasing anthropogenic Hg emissions during 2010-2015, suggesting that the measures taken during implementation of Minamata Convention may have reduce the anthropogenic emissions in recent years. Therefore, this study is valuable to atmospheric research communities and government. This paper is well organized and easily to be read. I have several comments that I wish could be addressed by the authors before the final publication of this paper.

1. line 34-37: this seems to be confusing. I would suggest the authors to separate these two periods throughout the manuscript because the trends during these two periods are quite different. As mentioned by the authors, the increasing trend was only observed between 2007 and 2014, and a declaration of an upward trend to 2017, to me, is at least not precise. If we look at the median (or mean) between 2007-2008 and 2016-2017, there is no clear increase of Hg concentrations.

   *We revised the text by pointing out that the 2007 – 2017 trend is driven mainly by a minimum in 2009 and maxima in 2014 and 2012. The 2007 – 2014 and 2012 – 2017 periods overlap and are thus difficult to separate.*

2. line 57-59: the study by Zhang et al., 2016 simulated the trend between 1990 and 2010 and compared them with field observations, that is not study related to the trend in last ten years. Also, I did not see any remarks from this study that anthropogenic Hg emissions is decreasing.

   *The text refers to the northern hemisphere and the trends of field observations shown in Figure 3 go until 2014 for North America and free troposphere. It thus covers a larger part of the 2007 – 2017 period and is relevant for our paper. The reviewer correctly mentions that the Zhang et al. (2016) do not find any substantial decrease of emissions between 2000 and 2010 which is at odds with the observed predominantly decreasing trends shown in Figure 3. The reasons for this discrepancy have still to be found.*

3. line 77-82: I would suggest to add Figure 1 to show the locations of the sampling sites. In some of following sections, the authors discussed the effect of atmospheric transport patterns on seasonal Hg trend. It will be excellent that the authors could provide the major air mass origins in wet and dry seasons.

   *Figure 1 with location of the sites is now added. Seasonal transport patterns at Cape Point are discussed in detail by Tshehla (2008).*

4. Line 133-144: From Figure 1, I see the seasonal variation in Hg at CPT (although not statistical significant) is opposite to that at AMS, with Hg peak in austral summer at CPT and in austral winter at AMS. I think the authors have not provide clear explanations for this. The contrast seasonal pattern indicate that the atmospheric transformation, foliar uptake, or oceanic

emissions might play an unimportant role here. The authors suggest that the frequent biomass burning in southern African and prevailing transport in austral winter may explain the seasonal trend at AMS. My question is why the enhanced biomass burning activities in southern African did not cause an increase of Hg in austral winter at CPT? which is located closer to southern African continent. Is the air mass transport from southern African to CPT less in austral winter than summer?

*We think that it does not make much sense to discuss phenomena which are not statistically significant. It is true that biomass burning in southern Africa peaks in austral winter and spring (Fig. 5 of Duncan et al., 2003) but most of the fires are located outside of the territory of South Africa and Namibia (Fig. 6 of Duncan et al., 2003) where most of the continental trajectories encountered at Cape Point originate (Tshehla, 2008).*

5. line 182-183: here the authors should indicate the sources of GEM. Are the sources related to oceanic emissions or long range transport from other countries or continents. This should be clarified here.

*We added a notice that oceanic sources are dominating. A long-range transport as an explanation is unlikely because of higher $CH_4$ and $CO_2$ concentrations at CPT in comparison with AMS while Hg concentrations are comparable. Both $CH_4$ and $CO_2$ have a longer lifetime than Hg and can thus serve as tracers for long-range transport.*

6. line 200-201: here the authors declare no clear GEM trend in 2012-2017. But in many other sections the authors also say a slightly downward trend. I think the authors should provide a consistent conclusion for the trend during this period.

*In the revised text we specified here that for the 2012 – 2017 period no significant trend was observed for Hg concentration averages and medians at AMS and for averages at CPT, and a slightly significant downward trend in medians at CPT. The calculated downward trends in averages and medians at CPT for the 2012 – 2017 period are comparable.*

7. line 277-279: it is very interesting that the authors explore the relationship between GEM trend and climate change (e.g., ENSO). But I am wondering that how would the climate effect the long-term trend of Hg. Will the occurrence of ENSO decrease or increase the natural Hg emissions from continents or oceans? I think it is better to provide a further discussion for this.

*Such discussion would need a complex model and a better knowledge of the mercury cycle. We note that there are still too large gaps in our knowledge which prevent a credible forecast of the climate change on future Hg emissions. The discrepancy between the observed trend and the emission inventories mentioned above in point 2 is just one example.*

Table 1-3: I think it is better to add the trend in percentile (the authors only provide absolute concentration trend) to these tables. This is because we cannot read whether the trend is strong or not based on concentrations. For example, does the annual increase of 0.05 mBq m$^{-3}$ yr$^{-1}$ of median Rn concentrations indicate a strong increase for Rn concentrations?

*All trends of $^{222}$Rn concentrations were insignificant and thus, in our opinion, it does not make much sense to discuss them anyway. The reviewer would probably like to express the trends in percent of the concentrations or mixing ratios. For $CO_2$ and $CH_4$ it is not the usual way of trend expression and as such its use would make it difficult to compare with published data. It also poses a question "percent of what" which is not always easy to answer. This is especially tricky for $^{222}$Rn concentrations which vary by more than a factor of 100 at CPT. We, therefore, stick with our presentation.*

Figure 2: I would suggest the authors add a figure to show the means of Hg. The authors used the means many times throughout the paper.

*As mentioned in the text the plot of annual averages and annual medians are almost identical and many points would merge if both averages and medians were plotted. If required, a table in Supporting Information can be added.*